# Primary Effusion Lymphoma: A Clinicopathologic Perspective

**DOI:** 10.3390/cancers14030722

**Published:** 2022-01-30

**Authors:** Diamone A. Gathers, Emily Galloway, Katalin Kelemen, Allison Rosenthal, Sarah E. Gibson, Javier Munoz

**Affiliations:** 1Department of Internal Medicine, University of New Mexico, Albuquerque, NM 87131, USA; 2Department of Medicine, University of Arizona College of Medicine, Phoenix, AZ 85721, USA; egalloway37@email.arizona.edu; 3Division of Hematopathology, Department of Laboratory Medicine and Pathology, Mayo Clinic, Phoenix, AZ 85054, USA; Kelemen.Katalin@mayo.edu (K.K.); Gibson.Sarah@mayo.edu (S.E.G.); 4Division of Hematology and Oncology, Mayo Clinic, Phoenix, AZ 85054, USA; Rosenthal.Allison@mayo.edu (A.R.); munoz.javier@mayo.edu (J.M.)

**Keywords:** primary effusion lymphoma, CAR-T therapy, HIV positive

## Abstract

**Simple Summary:**

Primary effusion lymphoma (PEL) is a rare B-cell lymphoma with a particularly aggressive course. This disease usually affects the variably immunocompromised and maintains a predilection for body cavities, leading to the development of malignant effusions. Pathogenesis suggests an association with HHV8 and EBV, although this is unclear. There are no definitive guidelines for the treatment of PEL, but commonly used initial regimens include those similar to DA-EPOCH and CHOP. The prognosis remains poor, even with advanced treatment. Newer therapies incorporate the use of axicabtagene ciloleucel, CAR-T cell therapies, and augmented EPOCH regimens. Further investigation into oncogenesis and targeted molecular pathways could provide insights into treatment targets. In this review, the authors would like to compare this disease with other similarly aggressive B-cell lymphomas, as well as highlight the unique epidemiology, pathogenesis, and current treatment options for patients diagnosed with PEL.

**Abstract:**

Primary effusion lymphoma (PEL) is a rare, aggressive B-cell lymphoma that usually localizes to serous body cavities to subsequently form effusions in the absence of a discrete mass. Although some tumors can develop in extracavitary locations, the areas most often affected include the peritoneum, pleural space, and the pericardium. PEL is associated with the presence of human herpesvirus 8 (HHV8), also called the Kaposi sarcoma-associated herpesvirus (KSHV), with some variability in transformation potential suggested by frequent coinfection with the Epstein-Barr virus (EBV) (~80%), although the nature of the oncogenesis is unclear. Most patients suffering with this disease are to some degree immunocompromised (e.g., Human immunodeficiency virus (HIV) infection or post-solid organ transplantation) and, even with aggressive treatment, prognosis remains poor. There is no definitive guideline for the treatment of PEL, although CHOP-like regimens (cyclophosphamide, doxorubicin, vincristine, and prednisone) are frequently prescribed and, given the rarity of this disease, therapeutic focus is being redirected to personalized and targeted approaches in the experimental realm. Current clinical trials include the combination of lenalidomide and rituximab into the EPOCH regimen and the treatment of individuals with relapsed/refractory EBV-associated disease with tabelecleucel.

## 1. Clinical Presentation and Epidemiology

PEL predominantly affects elderly or immunocompromised individuals, particularly those infected with HIV. This malignancy accounts for approximately 4% of all HIV-associated non-Hodgkin lymphoma (NHL) [1,2]. In studies of HIV-positive patients with PEL, the median age of diagnosis was 44–45 years old and 92–95% of the patients were male [3,4].

The clinical presentation of PEL is largely influenced by the location and extent of malignant lymphomatous effusions in body cavities. The classic form of PEL can affect the pleural, peritoneal, or pericardial space [1,5]. A patient with pleural effusions may present with shortness of breath. Those with pericardial effusions may have low blood pressure due to cardiac tamponade. General symptoms can include weight loss, fever, and night sweats [6]. B-cell lymphoproliferation may also lead to adenopathy or splenomegaly [3]. Common laboratory findings of PEL are compatible with severe immunosuppression and the depletion of CD4+ lymphocytes [7].

A variant of PEL may lead to development of the lymphoma in lymph nodes or as a solid tumor mass in extranodal sites such as the gastrointestinal tract, lung, skin, or central nervous system [4,8]. Although this extracavitary lymphoma is not associated with an effusion, it is still classified as a PEL variant due to its indistinguishable morphologic and immunophenotypic characteristics and association with HHV8 [9]. Guillet et al. studied 51 patients with HIV diagnosed with PEL and found that one third developed extracavitary solid tumors; these patients with extracavitary disease had similar demographics to those with classic PEL [4].

## 2. Pathogenesis and Association with Viral Infections and Immunosuppression (e.g., HIV, EBV, HHV8)

PEL was first described in the setting of the AIDS epidemic in the late 1980s and was later discovered to be associated with AIDS-associated Kaposi sarcoma [10,11]. A subsequent study identified HHV8 as the causative agent of PEL [5]. Approximately 80% of PEL cases, particularly those developing in patients with HIV, show coinfection with EBV [3,12]. However, the EBV in these cases shows restricted gene expression and is not thought to be required for the pathogenesis of this lymphoma [13,14,15]. In healthy individuals, the immune system limits HHV8-induced B-cell proliferation. However, when the immune system is impaired, as in the setting of HIV infection or iatrogenic immunosuppression, the viral promotion of cell proliferation may proceed unchecked, leading to HHV8-mediated lymphomagenesis (Figure 1) [16,17,18].

### Immune-Evasion Mechanisms by HHV8

The key for establishing a persistent HHV8 infection in the human cell is a complex interaction between HHV8 and the host immune system. HHV8 accomplishes this task by producing proteins homologous to human immune modulatory proteins that have likely been hijacked from human host cells over the course of viral evolution [14]. Persistent infections can be established by either a latent or a lytic viral life cycle. The latent life cycle results in a prolonged, stable, immunologically silent infection. During the lytic cycle, virions are released and infect new host cells [14,16]. HHV8 engages sophisticated cellular processes to make a persistent infection possible by interfering with the host immune responses.

A latent infection specifically targeting lymphoid cells is not unique to HHV8 but is a general feature of gamma herpesviruses [16]. Most lymphoproliferative disorders develop many years after a primary viral infection by a gamma herpesvirus, underlining the medical importance of the persistent latent infection. It is known that the lymphoma cells of PEL are in the latent phase of infection, a viral life cycle that is characterized by the lack of expression of most viral antigens [14,16]. While many other proteins are not expressed in latency, abundant expression of the latency-associated nuclear antigen (LANA) is crucial for the preservation of the viral genome in host cells during replication [18]. Interestingly, potential viral epitopes on LANA do not always provoke a strong response by the cytotoxic T-cells (CTLs), in part due to immune escape mechanisms [20,21]. One of the main immune-escape mechanisms of the latency phase is the inhibition of CTLs, a feat achieved by engaging a variety of proteins produced by HHV8, including viral caspase-8 and the (FLICE)-like inhibitory protein (vFLIP). Viral FLIP is a protein of strong anti-apoptotic properties in T-cell lines [22] and a promoter of nuclear factor kB (NF-kB) in host cells through engaging members of the TRAF family (tumor necrosis factor receptor-associated factor family) [23,24].

Many proteins produced by HHV8 target a central paradigm of T-cell mediated immune regulation, notably a promotion of the host T_H_2-cell responses and a concomitant downregulation of the T_H_1-dependent T-cell immunity. Several virally encoded cytokines, especially viral interleukin-6 (vIL-6), viral chemokines, and CD200, play a central role in this process. The pleiotropic cytokine IL-6 exhibits complex roles in human cells with important effects in hematopoiesis, inflammation, and oncogenesis by binding to one of two receptors, IL-6 receptor and gp130. HHV8 exploits these interactions by producing a homologue of IL-6 (vIL-6) that is encoded by the open-reading frame (ORF) K2 of the HHV8 genome. The expression of vIL-6 is well documented in both lytically infected and latently infected cells [25,26]. An important feature of vIL-6 is the capability to activate gp130 independently of the IL-6 receptor [27,28]. Once gp130 is activated, vIL-6 can initiate different pathways of immune evasion. The downstream effects of vIL-6 include the promotion of T_H_2-cell response over a T_H_1-type response [29] and the inhibition of the influx of neutrophils by increasing the expression of the chemokine CCL2 and decreasing the expression of CXCL8 [30].

Viral IL-6 is not the only human cytokine homologue produced by HHV8. Viral CC-chemokine ligand 1 (cCCL1), vCCL2, and vCCL3, which are also encoded by the HHV8 genome [25], target both T_H_2 cells and CD4+ CD25+ regulatory T cells [31]. ORF K14 of HHV8 encodes viral CD200 (vCD200), a glycosylated cell surface protein with a strong inhibitory effect on T_H_1-type cytokine production [32]. HHV8 also produces four different interferon (IFN)-regulatory factors, viral IRF1, vIRF2, vIRF3, and vIRF4, which are produced during the lytic reactivation and latency phase. As with their human homologues, these regulatory proteins inhibit type I interferon (IFN)-mediated signaling and IFN-mediated apoptosis [33].

In addition to its interference with the cell-mediated immunity, HHV8 also modulates the activity of the complement system. The complement system is capable of linking the innate and adaptive immune systems and plays an important role against viruses. HHV8 evades complement-mediated immunity by producing three complement lytic proteins (KCPs) that are encoded by ORF4 and are produced by alternative splicing [34]. KCPs contribute to the lysis of C3 convertase, the main product of the classical complement pathway, and they can inactivate C3b and C4b. Once the intact C3b and C4b are bound to the infected cell, they can exhibit a strong anti-viral effect by either lysing an infected cell or coating the viral particle and promoting complement-mediated phagocytosis [35]. The importance of this immune-evasion mechanism is that lymphoma genesis is highlighted by the observation of KCPs detected on the surface of the HHV8-infected cells of PEL [34].

Immune eradication of viral infections requires a strong CD8+ CTL-mediated attack on virus-infected cells. These cytotoxic cells are strongly dependent on MHC class I-dependent antigen presentation; therefore, they require a strong response by antigen presenting cells (APCs), such as dendritic cells, macrophages, and B cells. HHV8 engages several strategies to escape recognition by CD8+ CTLs. HHV8-infected cells express two homologous proteins, modulator of immune recognition 1 (MIR1) and MIR2, both capable of downregulating the expression of MHC class I molecules on the surface of infected cells [36,37,38]. These two proteins have been shown to be expressed in PEL and in latently infected B cells [39]. An additional effect of vMIR2 is the decreasing of the expression of CD86 and ICAM1, resulting in the inhibition of APCs [40,41].

In summary, HHV8 generates complex immune-regulatory mechanisms that antagonize both the innate and the adaptive immune system of the host, creating a milieu to promote HHV8-related lymphoproliferative disease. HHV8 encodes more than 10 homologues of cellular genes that are thought to play an important role in lymphomagenesis, including LANA, vIRF3, the viral homolog of cyclin D (vCYC), vFLIP, and vIL-6. NF-kB is constitutively activated in PEL and is required for cell survival [42]. Viral FLIP plays a major role in this NF-kB activation via both the classical and the alternative pathways, leading to the expression of both antiapoptotic and growth factor genes. The antiapoptotic functions of vFLIP have been shown to be critical for PEL tumor cell survival in vitro [16]. LANA additionally suppresses apoptosis by inhibiting TP53 and induces cell proliferation via the RB-E2F pathway [14]. Immune evasion also plays a role in the pathogenesis of PEL and is enhanced, at least in part, by vIRF3, which inhibits HLA trans-activators, resulting in reduced tumor cell recognition and killing by T cells [14,43].

## 3. Pathology and Ancillary Tests (e.g., HHV8, CD30, EBER, etc.)

The diagnosis of PEL may be made by the evaluation of a body fluid or a solid tissue mass in those cases with an extracavitary presentation. The neoplastic cells may be quite pleomorphic in body fluids, ranging from large immunoblastic/plasmablastic cells to more anaplastic-appearing cells (Figure 2) [1,14,44,45,46]. The lymphoma cells contain large nuclei with round to irregular nuclear contours and prominent nucleoli and generally contain abundant basophilic cytoplasm. Some cells may show a perinuclear “hof” or halo compatible with plasmacytoid differentiation. Occasional multinucleated Reed-Sternberg-like cells may be present. Although the morphologic features of extracavitary PELs are similar, the neoplastic cells may have a more uniform appearance than those found in body fluids, and such cases may show the involvement of lymph node sinuses or other lymphatic or vascular channels mimicking anaplastic large-cell lymphoma or intravascular lymphoma [44,45].

The neoplastic cells of PEL originate from post-germinal center B cells that show further plasmablastic differentiation [14,44]. The cells usually express the leukocyte common antigen CD45 and show variable staining for plasma-cell-associated markers, such as CD138, CD38, IRF4/MUM1, and EMA [1,14,44,45,46]. The neoplastic cells generally lack expression of the pan-B-cell markers CD19, CD20, CD79a, and PAX5, and the surface and cytoplasmic immunoglobulin are usually negative or low in expression. However, it is noted that the extracavitary variant of PEL shows a slightly more frequent expression of B-cell-associated markers and immunoglobulins compared to classic PEL [44,45,46]. Although T-cell- and natural-killer-cell-associated antigens are generally negative, aberrant expressions of some T-cell markers, including CD3, may occasionally be seen, particularly in those cases with extracavitary presentation [14,44,45,46]. CD30, which is frequently expressed by PELs, raises the differential diagnosis of both anaplastic large-cell lymphoma and classic Hodgkin lymphoma [14,44,45,46]. However, the diagnosis of PEL can be confirmed with immunohistochemical staining for LANA1, an HHV8-associated latent protein, which is positive in the nuclei of the PEL cells [6,14,44,45,46,47]. EBV may be demonstrated with in situ hybridization for EBV-encoded small RNA (EBER) in many cases, while EBV LMP1 is absent [44,46].

PEL shows clonally rearranged and hypermutated immunoglobulin genes, and some cases may show genotypic infidelity by harboring a clonal T-cell receptor gene rearrangement in addition to a clonal immunoglobulin gene rearrangement [1,44,45]. Gene expression profiling of AIDS-related PEL supports its plasmablastic derivation, with a distinct profile combining features of plasma cells and EBV-transformed lymphoblastoid cell lines (4,44). Although PELs may show complex karyotypes, no recurrent chromosomal alterations have been described [1,44,45,46]. Mutations of the *BCL6* gene are seen in a subset of cases [44,48].

## 4. Differential Diagnosis (HHV8-Negative Effusion-Based Lymphoma, Pyothorax-Associated Lymphoma, Burkitt Lymphoma, Diffuse Large B-Cell Lymphoma, Not Otherwise Specified, Plasmablastic Lymphoma, Anaplastic Large-Cell Lymphoma, Poorly Differentiated Carcinoma)

Lymphomas presenting as effusions without an associated tumor mass are rare [45]. Aside from PEL, there are other lymphomas that are not associated with HHV8 that could be considered in the differential diagnosis (Table 1). Effusion-based lymphomas that morphologically resemble PEL, but are not associated with HHV8 or HIV infection, are described in older individuals with underlying medical conditions leading to fluid overload. These lymphomas commonly express pan-B-cell markers and a subset are EBV-positive and may show abnormalities of *MYC*. HHV8-negative effusion-based lymphomas have a more favorable prognosis compared to PEL [19,45,49].

Diffuse large B-cell lymphomas (DLBCL) associated with chronic inflammation may occur in body cavities, with pyothorax-associated lymphoma (PAL) being the prototypical form [44,45,50]. PAL develops in the pleural cavity of patients with longstanding pyothorax and is more frequently associated with a local solid infiltrate [44,45,50]. Although PAL is also composed of B cells that may have plasmacytoid differentiation, these lymphomas are associated with EBV and not HHV8 [44,45,50,51].

Other B-cell lymphomas may occasionally involve serous fluids, including Burkitt lymphoma (BL) and DLBCL, not otherwise specified (NOS) [44,45,52]. BL, particularly the variant with the plasmacytoid differentiation that is more commonly seen in patients with HIV, may have cytomorphologic overlap with PEL [44,45]. However, BL is not associated with HHV8 and may be easily distinguished from PEL by its pan-B-cell marker expression, characteristic staining for CD10 and BCL6, and the presence of an *MYC* gene rearrangement [44,45]. The cytomorphologic features of the immunoblastic variant of DLBCL, NOS may also overlap with PEL [44]. However, the majority of DLBCL, NOS involving serous fluids are associated with contiguous or disseminated disease [52]. As with BL, DLBCL may be distinguished from PEL by the expression of pan-B-cell markers and the lack of association with HHV8. It is also important to note that compared to DLBCL, PEL has a higher degree of association with hypoalbuminemia, thrombocytopenia, and elevated IL-10 levels [3]. In addition, the cytomorphologic and immunophenotypic features of PEL overlap with those of plasmablastic lymphoma (PBL), particularly in cases of extracavitary PEL [44,45]. PBL also occurs in association with HIV infection or other immunodeficiencies but more frequently presents as a solid mass in extranodal regions of the head and neck or gastrointestinal tract and is associated with EBV and not HHV8 [44,45,53].

The cytomorphology and frequent CD30 expression seen in PEL raises the differential diagnosis of anaplastic large-cell lymphoma (ALCL) [20,45,46]. Occasional PELs may show some T-cell marker expression, which makes distinguishing these lymphomas even more difficult [14,44,45,46]. However, ALCL only occasionally involves serous body cavities, expresses ALK protein in most cases, and is not associated with HHV8 [44,45]. A poorly differentiated carcinoma is also in the differential diagnosis of PEL, particularly in extracavitary cases that show infiltration of sinusoids or vascular spaces [45]. However, expression of CD45 and lack of cytokeratin staining in PEL would help to exclude carcinoma. As can be seen, the diagnosis of PEL requires a multiparameter approach with extensive immunophenotyping and confirmation of HHV8 infection in all cases [44,45].

## 5. Prognosis and Treatment

There is no definitive guideline for the treatment of PEL, and even with treatment, prognosis remains poor. Median survival with treatment in some studies remains less than 24 months [3]. Prognosis is worsened by performance status, concurrent untreated HIV, and the involvement of the pericardial cavity as opposed to the pleural or peritoneal cavities. In addition, EBV positivity in patients with PEL has been associated with improved survival, in contrast with the association of elevated IL-6 levels with inferior survival [3]. It is suggested that disease involvement of greater than one body cavity is an adverse prognostic indicator; Narkhede et al. found that clinically significant overall survival decreased from 18 months to 4 months in patients with single versus multiple cavity involvement, respectively [6,54,55].

### 5.1. Current Treatment Options

Common regimens for treatment of PEL include dose-adjusted (DA) EPOCH (etoposide, prednisone, vincristine, cyclophosphamide, and doxorubicin) or traditional chemotherapy with CHOP (cyclophosphamide, doxorubicin, vincristine, and prednisone), which are typically used for treatment of other aggressive lymphomas [10,55]. CHOP is a commonly prescribed regimen for NHL, but it has not been shown to lengthen median survival for PEL [55]. Guillet et al. found, in a single-center retrospective analysis of 45 of 51 patients diagnosed with PEL, that treatment with a CHOP-derived regimen achieved complete remission (CR) in 62% of the classic (cavitary) PEL group with a 71.5% 2-year disease free survival rate [4].

EPOCH infusion therapy is a well-tolerated regimen that has been modified with success to DA-EPOCH [56]. DA-EPOCH incorporates the use of platelet and absolute neutrophil counts to guide subsequent dose adjustments to etoposide, doxorubicin, and cyclophosphamide [57]. It has been suggested that this responsive dose adjustment can overcome limitations encountered by traditional EPOCH therapy, ultimately leading to improved outcomes in comparison with more traditional regimens [58].

Rituximab has been explored for use in CD20+ PEL, but this marker is infrequently expressed by PEL. The addition of high-dose methotrexate (HD-MTX) to the CHOP regimen has shown a modest increase in median overall survival (OS) from 6 to 10 months but HD-MTX toxicity related to its tendency to accumulate in effusions is an undesirable effect of use [14]. Talc pleurodesis, often used for the management of recurrent malignant pleural effusions, could also be considered as an option for potentially alleviating symptoms [59]. Many of these individual therapies have been assessed for palliative use as independent agents in the elderly and those with poor functional status felt not to be able to tolerate more aggressive therapies.

### 5.2. Relapsed and Refractory PEL

The National Comprehensive Cancer Network (NCCN) B-cell Lymphoma Guidelines (Version 3.2021) include axicabtagene ciloleucel (Axi-cel) as a category 2A recommendation for the treatment of PEL, relapsed/refractory DLBCL, HHV8-positive DLBCL, and similar conditions for patients with relapsed/refractory disease [60,61]. Autologous stem cell transplant in combination with high-dose chemotherapy has been used to achieve a CR period of 12 months [62]. Cassoni et al. showed that a similar period of remission was achieved using radiotherapy directed towards localized pleural masses in a patient diagnosed with chemotherapy-refractory HIV-associated PEL [63]. The use of ibrutinib, lenalidomide, and rituximab in the treatment of similarly aggressive relapsed/refractory non-germinal center B-cell-like DLBCL has shown promising activity with an objective response rate (ORR) of 65% and 41% with CR, although this study did not include patients with PEL [64]. In a small cohort, the immune checkpoint inhibitor pembrolizumab (in various combinations with pomalidomide) showed an OS of about 15 months when used to treat patients living with HIV (PLWH) who have relapsed/refractory disease [65]. A case study by Marquet et al. showed radiological CR and transformation of HHV8 DNA from positive to negative after the initiation of treatment with oral valganciclovir [66].

### 5.3. HIV-Positive PEL

Consideration must be given to HIV status, as many patients diagnosed with PEL are HIV positive. Ramaswami and team highlight that PEL associated with the often-underdiagnosed multicentral Castleman’s disease led to a clinically significant reduction in overall survival from 71% at 10 years to a 5-year OS of 38% [67]. Patients with AIDS-NHL should be treated with full-dose chemotherapy regimens and concurrent antiretroviral therapy (ART) in a multidisciplinary team inclusive of a hematologist and an HIV specialist. As lack of anti-HIV therapy is a poor prognostic indicator of PEL, the concurrent combination ART (cART) has been shown to allow for faster immune recovery (although cART therapy does not directly influence OS) when used in combination with chemotherapy [57]. It has been suggested that if ART has not already been initiated at the time of the lymphoma diagnosis, determining the sequence of initiation of ART and chemotherapy should be on a case-by-case basis. There should be special care taken to monitor for the development of cytopenias and drug–drug interactions in this population. Known interactions with ART include vinca alkaloid and taxane toxicity with protease inhibitors and cobicistat, severe cytopenias and neuropathy with nucleotide reverse transcriptase inhibitors, and a decrease in vinca alkaloid and taxane levels with non-nucleoside reverse-transcriptase inhibitors due to induction of the CYP3A4 enzyme. To avoid drug interactions, it is advisable that the ART regimen be modified when treatment for PEL is initiated. Antiviral therapies such as cidofovir and interferon have also been used. Uldrick et al. found that in HIV patients with concurrent diagnoses of non-AIDS-defining cancers who were also on ART, pembrolizumab had an acceptable safety profile with persistent HIV control and stable CD4 counts [68]. In all patients with HIV being treated for lymphoma, it is necessary to provide G-CSF support and Pneumocystis jiroveci pneumonia (PJP) prophylaxis and prevent tumor lysis syndrome. There is an increased risk for concomitant opportunistic infections and usually a need to include central nervous system chemoprophylaxis.

### 5.4. Overlap of PEL with Kaposi’s Sarcoma (KS) and Kaposi Sarcoma-Associated Herpesvirus Multicentric Castleman Disease (KSHV-MCD)

Kaposi sarcoma-associated herpesvirus (HHV8) is directly associated with the development of concurrent PEL and/or Multicentric Castleman Disease. This is likely due to a common viral etiology and KSHV’s ability to utilize B cells as a persistent reservoir of inflammatory cytokines, leading to proliferation and transformation to PEL and MCD. Interestingly, EBV-positive status in PEL is associated with improved survival compared to the more commonly occurring HIV-DLBCL. However, higher-than-normal circulating inflammatory markers (specifically IL-6, IL-10, and viral IL-6), as well as increased HHV8 viral loads, are often found in patients with concurrent PEL and KSHV-MCD, which confer a worse prognosis [69,70]. NCCN guidelines recommend treatment of KSHV-MCD with rituximab-based regimens; however, it is recommended that patients with concurrent PEL receive cytotoxic chemotherapy in addition to these rituximab-based regimens [3,67]. Rituximab has been shown to downregulate inflammatory cytokines as well as work synergistically with common topoisomerase inhibitors and other antineoplastic agents [71,72]. The effective use of rituximab, regardless of the CD20 status in patients with concurrent PEL and KSHV-MCD, was evidenced by the work of Lurain et al. who suggest that the use of rituximab could decrease the likelihood of patients diagnosed with KSHV-MCV developing PEL, given its downregulatory effects on inflammatory cytokines [69].

### 5.5. CAR-T Therapies in Other B-Cell Lymphoproliferative Disorders

Axicabtagene ciloleucel (Axi-cel) is a chimeric antigen receptor (CAR) T-Cell agent, currently used as a third-line treatment for follicular and large B-cell lymphoma. ZUMA-1 is a phase 2 clinical trial across multiple centers, including 111 patients. Axi-cel was successfully manufactured for 99% of these patients and administered to 91%, resulting in an overall response rate (ORR) of 82% and a complete response rate (CRR) of 54%. These responses were sustained with a median follow-up of about 15 months and an OS at 18 months of 52% [73]. Brexucabtagene autoleucel (previously KTE-X19) is another CAR-T cell therapy, typically used for relapsed/refractory mantle cell lymphoma. This therapy was used in the 74-patient ZUMA-2 trial, ultimately showing a 12-month progression free survival (PFS) of 61% and OS of 83%. The most common adverse events were those effects previously experienced after CAR-T therapy, including cytopenias (94%) and infections (32%) [74]. Similar success was observed in the smaller ATTCK-20-2 trial, which incorporated the use of ACTR087 plus rituximab in patients with relapsed/refractory CD20-positive B-cell lymphoma. Of the 26 patients enrolled, 69% were diagnosed with relapsed/refractory DLBCL and subsequently treated with fixed doses of rituximab administered concurrently with variable doses of ACTR087, leading to an ORR of 50%. In addition, the team was able to further define safety monitoring and adverse reaction management guidance across clinical studies inclusive of ACTR T-cell products [75]. That said, the pivotal trials that led to the FDA approval of CAR-T cell therapies excluded patients with HIV infection.

In addition to these trials, case reports have shown promise in the use of CAR-T cell therapies for patients with DLBCL in the setting of HIV infection. Abbasi et al. found that treatment of relapsed/refractory lymphoma with axi-cel led to significant responses with 8/10 patients, including 2 with prior CNS involvement, achieving a complete response at 3 months and manageable toxicities [76]. Similarly, Abramson et al. treated two patients with refractory high-grade B-cell lymphoma in the setting of HIV; one with grade 2 cytokine release syndrome but with achievement of the CR of the disease within one month, which was sustained for at least one year, and the other with CR in 28 days [77]. Recommended steps to optimize success of CAR-T therapy in the HIV+ patient include assessment of the HIV control and T-cell repertoire, infection control with a recommended absolute lymphocyte count of >100 for the use of successful CAR-T manufacturing, and assessment of the drug/drug interactions to minimize overlapping toxicities between CAR-T and ART. In the post CAR-T therapy phase, it is critical to monitor HIV control, assess immune reconstitution and determine the need for G-CSF as well as maintain appropriate infectious prophylaxis, including coverage for PJP, fungal infections, HSV, and VZV [78].

Future opportunities for treatment of PEL include therapies targeting the mTOR pathway, the ribonucleotide reductase signaling pathway, CD30, and proteosome inhibitors as well as a focus on CD38 as a specific marker of PEL cells via the use of daratumumab, which may induce tumor response as well as decreased HHV-8 levels [79,80,81,82,83,84]. The immune-escape phenomenon is one plausible explanation for the challenges in treatment of PEL; therefore, several molecular-targeted therapies, including NF-kB, JAK/STAT, and PT3-kinase/AKT pathways are under investigation to treat PEL [14].

### 5.6. Clinical Trials

A prospective phase I-II trial is underway at the National Cancer Institute (NCT02911142) to evaluate the efficacy of an expanded EPOCH regimen to include rituximab and the immunomodulator lenalidomide (EPOCH-R2) in the treatment of PEL [69,85]. Participants were prescribed EPOCH and rituximab with the addition of a set dose of lenalidomide, administered for days 1–10 at the beginning of each 21-day cycle (total of 6). Preliminary results have shown promise, with a 2-year OS calculated at 66.7%, pending phase II outcomes [69]. The ongoing phase II trial NCT04554914 is a multicenter, multicohort trial set to evaluate treatment of individuals with relapsed/refractory EBV-associated disease with tabelecleucel which will be administered in 35-day cycles at a dose of 2 × 10^6^ cells/kg intravenously (IV), weekly for 3 weeks, and assessed for response [86].

## 6. Summary/Conclusions

Primary effusion lymphoma is a rare, aggressive disease with a very poor prognosis. Further investigation into oncogenesis and targeted molecular pathways could provide great insights into treatment targets. Guidance for current treatment regimens is largely based on limited case reports and expert consensus, and treatment choice requires careful consideration of comorbidities, performance status, potential medication interactions, and the presence of concurrent HIV infection. Clinical trials are in the early stages but to date provide promising potential for progress in this uncommon disease.

## Figures and Tables

**Figure 1 cancers-14-00722-f001:**
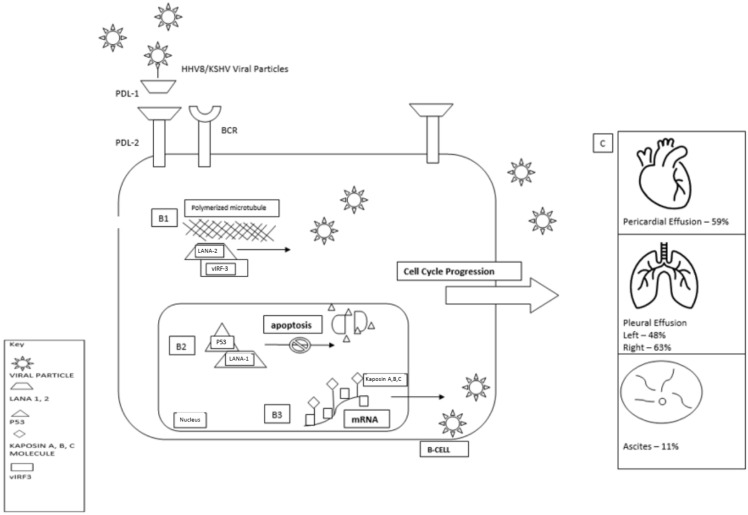
Pathogenesis Overview of PEL based on references [14,19]. P53: Tumor suppressor protein/transcription factor. Viral particle: Invasive HHV8, example of cell entry accessed via PDL-1. LANA-1,2: Latency associated nuclear antigen/oncoprotein. PD 1, 2: B cell surface marker. Kaposin molecule: representative of Kaposin protein A, B or C. A: Viral invasion of B-cell via PD-1/PDL-1 pathway. B1: Binding of LANA-2/vIRF3 complex to polymerized microtubules leads to transcription instability, facilitates viral proliferation. B2: Binding of LANA-1 to P53 prevent cell apoptosis facilitates viral proliferation. B3: Binding of Kaposi A, B, or C to variable sites of mRNA facilitates viral proliferation. C: Predominance of post-infectious effusion by organ site.

**Figure 2 cancers-14-00722-f002:**
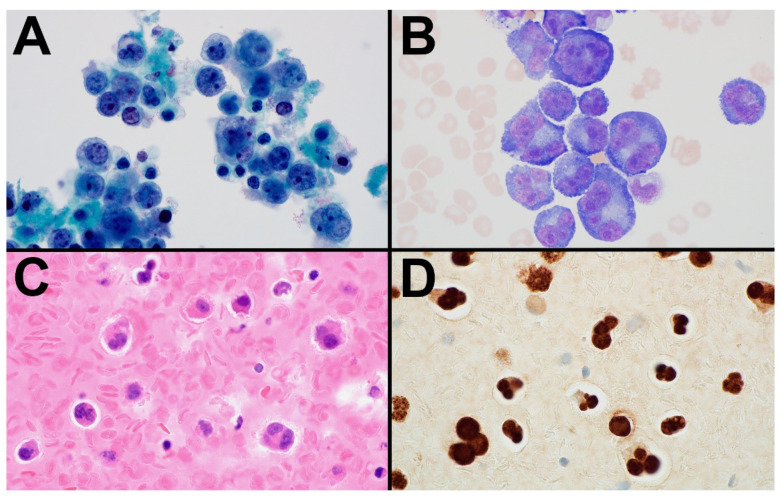
PEL arising in a patient who had previously received an allogeneic hematopoietic stem transplant for T-cell lymphoma. The cytospin slides (**A**,**B**) and cell block (**C**) of the pleural fluid in this case show a proliferation of large, pleomorphic lymphoid cells. Although some neoplastic cells have relatively round nuclear contours with prominent nucleoli, many others have a more anaplastic appearance with multinucleated forms that resemble Hodgkin/Reed-Sternberg cells. The neoplastic cells express CD138, EMA, and CD30 (Gibson, S.E., Division of Hematopathology, Department of Laboratory Medicine and Pathology, Mayo Clinic, Phoenix, AZ 85054, USA; Gibson.Sarah@mayo.edu (S.E.G.), 2021) and are strongly positive for HHV8 (**D**), confirming the diagnosis of PEL. (**A**) Papanicolaou stain; (**B**) Wright stain; (**C**) H&E stain; and (**D**) immunohistochemical stain with hematoxylin counterstain; (**A**–**D**) magnification × 1000).

**Table 1 cancers-14-00722-t001:** Differential Diagnosis of Primary Effusion Lymphoma.

	Primary Effusion Lymphoma	HHV8-Negative Effusion-Based Lymphoma	Pyothorax-Associated Lymphoma	Plasmablastic Lymphoma	Burkitt Lymphoma	Diffuse Large B-Cell Lymphoma *
Clinical presentation	Pleural, peritoneal, or pericardial effusion +/− nodal/extranodal mass B symptoms	Pleural, peritoneal, or pericardial effusion Fluid overload	Chest pain, fever, cough, dyspnea, chest wall mass History of TB related pyothorax	Extranodal tumor of head and neck or other extranodal sites +/− nodal disease B symptoms, variable based on immunologic status	Facial, abdominal, or other extranodal mass +/− nodal diseaseRapid tumor growth	Nodal or extranodal mass B symptoms in one-third
Morphology	Large, pleomorphic immunoblastic, plasmablastic or anaplastic cells	Large, pleomorphic immunoblastic or plasmablastic cells	Diffuse proliferation of centroblastic or immunoblastic cells, may show plasmacytoid differentiation	Diffuse proliferation of large, immunoblastic or plasmablastic cells	Medium-sized, monotonous cells with frequent cytoplasmic vacuoles; rarely resemble small, plasmacytoid immunoblasts	Diffuse proliferation of large immunoblastic cells (round nuclei and a single prominent nucleolus), may include cells with plasmacytoid features
EBV positivity	70%	30%	70%	60–75%	20–40% (100% in endemic form)	9–15%
HHV8 status	+	-	-	-	-	-
Phenotype	CD45+ CD20− CD19− CD79a− PAX5− IRF4/MUM1+/− CD38+/− CD138+/− CD30+/− Ig light chain−/+	CD45+/− CD20+/− CD19+/− CD79a+/− PAX5+/− IRF4/MUM1+/− CD38−/+ CD138−/+ CD30−/+ Ig light chain+/−	CD45+ CD20+/− CD19+ CD79a+/− PAX5+ IRF4/MUM1+ CD138−/+ CD30+/− Ig light chain+/−	CD45−/+ CD20− CD19− CD79a−/+ PAX5− IRF4/MUM1+ CD38+ CD138+ CD30−/+ Ig light chain+	CD45+ CD20+ CD19+ CD79a+ PAX5+ IRF4/MUM1−/+ CD38+ CD138− MYC+ CD10+	CD45+ CD20+ CD19+ CD79a+ PAX5+ IRF4/MUM1+/− CD38−/+ CD138−/+ CD30−/+
Cellular origin	Post-GC B cell with plasmablastic differentiation	GC or post-GC B cell	Post-GC B cell	Plasmablast	GC B cell	GC or post-GC B cell
HIV status	+++/−	−−−/+	-	++/−	−−/+	−−−/+
Other associations	Organ transplant; elderly	Fluid overload	Long-standing pyothorax	Organ transplant; other iatrogenic immunodeficiency; elderly	Malaria (endemic form); organ transplant; primary immune disorders	Organ transplant; other iatrogenic immunodeficiency; primary immune disorders
Prognosis	Median survival < 2 yr	2-yr survival 85%	5-yr survival 20–35%	Median survival <1 yr	Variable 5-yr survival ≥70%	Variable 5-yr survival >60%
Anatomic site	Body cavities (rare extracavitary)	Body cavities	Thoracic cavity	Extranodal (<10% nodal)	Extranodal (less frequent nodal)	Nodal or extranodal

PEL = Primary Effusion Lymphoma. DLBCL = Diffuse Large B-cell Lymphoma. PAL = Pyothorax associated Lymphoma. GC = Germinal center. LAD = Lymphadenopathy. TB = Tuberculosis. CNS = Central Nervous System. GI = Gastrointestinal. B Symptoms = constitutional symptoms including fevers, night sweats, and weight loss, typically associated with lymphomas. EBV, Epstein-Barr virus; GC, germinal center; HHV8, human herpes virus 8; HIV, human immunodeficiency virus; TB, tuberculosis; yr, year; * immunoblastic variant.

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
