# Peer review of "Primary Effusion Lymphoma: A Clinicopathologic Perspective"

_cancers, 2022, doi:10.3390/cancers14030722_

Round 1

Reviewer 1 Report

It is a well-written and comprehensive review covering all the significant aspects of PEL.

Please be more descriptive in the legend of Figure 1.

Author Response

It is a well-written and comprehensive review covering all the significant aspects of PEL.

Please be more descriptive in the legend of Figure 1.

Response: Thank you, we agree that the image description could be more robust and would like to adjust the descriptions of the key terms to include the following:

TP53: Tumor suppressor protein/transcription factor for cell cycle arrest/apoptosis

Virus: invasive HHV8, example of cell entry accessed via PDL-1

Viral Protein: product of viral invasion, replicated by cell machinery

LANA-1: latency associated nuclear antigen/oncoprotein, necessary for cell cycle progression/apoptosis

LANA-2: latency associated nuclear antigen/oncoprotein, necessary for cell cycle progression/apoptosis

Reviewer 2 Report

I congratulate the authors on the elaboration of the present paper in a logical and comprehensible manner to the reader, offering a documented global approach regarding a subject that is less common.

I would recommend a minor revision of the English language and spelling errors.

Author Response

I congratulate the authors on the elaboration of the present paper in a logical and comprehensible manner to the reader, offering a documented global approach regarding a subject that is less common.

I would recommend a minor revision of the English language and spelling errors.

Response: We appreciate your attention to this detail and thank you for your compliment – in response, we have repeated a manual and electronic review of grammar/spelling errors and corrected these as appropriate.

Reviewer 3 Report

This is a well-written review of the clinical presentation, pathogenesis, epidemiology, and current treatment landscape for primary effusion lymphoma (PEL), a rare, aggressive, B-cell lymphoma. However, the authors do not provide enough distinction between PEL and other aggressive B-cell lymphomas that have a similar clinical and/or pathologic features. In addition, I think that the authors could do more in discussing the differential diagnosis and prognosis (e.g. viral and immunologic differences between PEL and DLBCL). Moreover, there is need for more detail on the biological rationale for using rituximab (including clarification that it is being used successfully in CD20 negative PEL) and the concurrent presentation of PEL with KSHV-associated multicentric Castleman disease (KSHV-MCD) and Kaposi sarcoma (KS). Finally, I would like to see a more extensive discussion on current and future treatment options that includes the use of immunotherapy and immunomodulatory drugs rather than the evidence that the authors present.

Specific comments:

In comparing the features of aggressive B-cell lymphomas (Table 1), the authors should make the following changes:

  1. Provide a footnote to explain "B symptoms" to make the article more accessible
  2. EBV positivity is currently ascribed +/- but would be more informative if presented on a continuous scale e.g. 80% for PEL, 100% for pyothorax associated lymphoma etc (with appropriate references)
  3. Immunophenotype is also currently presented as +/-, yet this varies widely by disease and has important considerations for treatment e.g. PEL and plasmablastic lymphoma are highly CD38+ (bright); would recommend a scale of 0 to +3 with 0 being no expression and +3 being bright/high expression for each of the cell surface markers.
  4. HIV status is also currently +/- suggesting, for example, that PEL and DLBCL have similar association with HIV, yet PEL is predominantly found in HIV positive individuals. Suggest using a proportion of patients who have HIV to more accurately depict the association.
  5. Median survival for PEL is stated as <1 year. In the paper by Lurain et al, median survival was 22 months [Lurain K et al. Viral, immunologic, and clinical features of primary effusion lymphoma. Blood. 2019 Apr 18;133(16):1753-1761. doi: 10.1182/blood-2019-01-893339. Epub 2019 Feb 19. PMID: 30782610; PMCID: PMC6473499.]

Suggest providing a range of median survival to reflect these more recent improvements in outcomes. Both in table 1 and in section 5 on Prognosis and treatment (line 240).

  1. Suggest adding another row to show association with other immunosuppressive conditions in HIV-negative individuals, especially for PEL and PBL.

In discussing the differential diagnosis of PEL in patients with HIV, it would be important to include that compared to DLBCL, PEL is significantly associated with hypoalbuminemia, thrombocytopenia, and elevated IL-10 levels [Lurain K et al. Viral, immunologic, and clinical features of primary effusion lymphoma. Blood. 2019 Apr 18;133(16):1753-1761. doi: 10.1182/blood-2019-01-893339. Epub 2019 Feb 19. PMID: 30782610; PMCID: PMC6473499.]

In discussing prognosis, would also add that among patients with PEL, EBV positivity is associated with improved survival whereas elevated IL-6 levels are associated with inferior survival [Lurain K et al. Viral, immunologic, and clinical features of primary effusion lymphoma. Blood. 2019 Apr 18;133(16):1753-1761. doi: 10.1182/blood-2019-01-893339. Epub 2019 Feb 19. PMID: 30782610; PMCID: PMC6473499.]. Additionally, would also explain that with concurrent MCD the presence of PEL and KS, would impact the overall survival of patients (Ramaswami R, Lurain K, https://ashpublications.org/bloodadvances/article/5/6/1660/475480/Characteristics-and-outcomes-of-KSHV-associated).

In discussing treatment options, the authors mention the use of rituximab but mention that it has been explored for use in CD20+ PEL. However, rituximab has been used in PEL regardless of cell surface expression of CD20. Suggest that the reviewers expand the discussion of use of rituximab and explore the rationale in greater detail. For example, in the study of EPOCH-R2 for PEL, investigators justify the inclusion of rituximab by applying the current understanding of the pathobiology of IL-6 syndromes in KSHV-MCD and KSHV-NHL. Specifically, rituximab is effective in KSHV-MCD which is CD20 negative; PEL patients have similar clinical symptoms, suggesting a common pathophysiology likely driven by KSHV-infected B cells associated with increases in inflammatory cytokines (IL-6, IL-10, and viral IL-6). It is hypothesized that by depleting depletes B cells, Rituximab may therefore reduce levels of inflammatory cytokines that would otherwise lead to the clinical features of PEL [REF: Lurain K, Ramaswami R et al. Phase I/II Study of Lenalidomide Combined with DA-EPOCH and Rituximab (DA-EPOCH-R2) in Primary Effusion Lymphoma in Patients with or with-out HIV. Blood. 2019;134(Supplement_1):4096-4096. doi:10.1182/blood-2019-122070].  Moreover, the authors fail to mention the common overlap of PEL with KS and KSHV-MCD. These overlap syndromes are distinct entities which call for more nuanced treatment approaches e.g. addition of rituximab in PEL/KSHV-MCD overlap (as noted above).

The authors also fail to include the emerging use of immunotherapy in HIV positive patients with cancer, including non-Hodgkin lymphoma. In the study by Uldrick et al. [Uldrick TS, Gonçalves PH et al. Assessment of the Safety of Pembrolizumab in Patients With HIV and Advanced Cancer-A Phase 1 Study. JAMA Oncol. 2019 Sep 1;5(9):1332-1339. doi: 10.1001/jamaoncol.2019.2244. PMID: 31154457; PMCID: PMC6547135], investigators showed that the use of pembrolizumab is safe and has clinically beneficial activity against PEL.

The authors also do not mention the use of immunotherapy combined with immunomodulatory drugs as a viable treatment option for patients with relapsed/refractory PEL. In another study from the NCI, the use of pembrolizumab with or without pomalidomide also showed responses/disease control in patients with PEL [Lurain K, Ramaswami R, et al. Use of pembrolizumab with or without pomalidomide in HIV-associated non-Hodgkin's lymphoma. J Immunother Cancer. 2021 Feb;9(2):e002097. doi: 10.1136/jitc-2020-002097. PMID: 33608378; PMCID: PMC7898875.]

Finally, in discussing future opportunities for treatment (line 339), the authors fail to mention targeting CD38, a common marker of PEL cells. Shah et al. successfully treated a patient with PEL using daratumumab, an anti-CD38 monoclonal antibody [Shah NN et al. Daratumumab in Primary Effusion Lymphoma. N Engl J Med. 2018 Aug 16;379(7):689-690. doi: 10.1056/NEJMc1806295. PMID: 30110586.]

Author Response

This is a well-written review of the clinical presentation, pathogenesis, epidemiology, and current treatment landscape for primary effusion lymphoma (PEL), a rare, aggressive, B-cell lymphoma. However, the authors do not provide enough distinction between PEL and other aggressive B-cell lymphomas that have a similar clinical and/or pathologic features. In addition, I think that the authors could do more in discussing the differential diagnosis and prognosis (e.g., viral and immunologic differences between PEL and DLBCL). Moreover, there is need for more detail on the biological rationale for using rituximab (including clarification that it is being used successfully in CD20 negative PEL) and the concurrent presentation of PEL with KSHV-associated multicentric Castleman disease (KSHV-MCD) and Kaposi sarcoma (KS). Finally, I would like to see a more extensive discussion on current and future treatment options that includes the use of immunotherapy and immunomodulatory drugs rather than the evidence that the authors present.

Specific comments:

In comparing the features of aggressive B-cell lymphomas (Table 1), the authors should make the following changes:

  1. Provide a footnote to explain "B symptoms" to make the article more accessible

                Response: Thank you, the footnote beneath Table 1 has been expanded to include more descriptive language.

  1. EBV positivity is currently ascribed +/- but would be more informative if presented on a continuous scale e.g. 80% for PEL, 100% for pyothorax associated lymphoma etc (with appropriate references)

Response: Thank you. We agree with your assessment and have updated Table 1 with representative percentages for EBV and Burkitt lymphoma in an attempt to be inclusive of the degree of variability based on the study, nature of disease being endemic vs sporadic/immunodeficiency associated and geographic location).

  1. Immunophenotype is also currently presented as +/-, yet this varies widely by disease and has important considerations for treatment e.g., PEL and plasmablastic lymphoma are highly CD38+ (bright); would recommend a scale of 0 to +3 with 0 being no expression and +3 being bright/high expression for each of the cell surface markers.

Response: We greatly appreciate your assertion and wholeheartedly agree with your suggestion, although we would like to respectfully propose that given the great degree of variability in expression, assigning at 0-3+ scale may be lead to some misunderstanding across references for some of our potential readers, partially evidenced by the conundrum of being unable to find multiple references that were in agreement in offering consistent percentages of expression that would represent this data most effectively. If we can improve upon how this data is currently represented, we would be happy to modify this table in a way that you feel is most appropriate. Thank you.

  1. HIV status is also currently +/- suggesting, for example, that PEL and DLBCL have similar association with HIV, yet PEL is predominantly found in HIV positive individuals. Suggest using a proportion of patients who have HIV to more accurately depict the association.

Response: Thank you for this advice. Similar to the challenge in representing immunophenotype by percentage, we found that there were not reliably concordant references to extrapolate exact proportions of patients with HIV to better illuminate this particular representation. In order to incorporate your revision and hopefully improve upon what we initially documented, we modified the HIV status section of the table to help with a more visually represented quantitative explanation for our potential readers. Please let us know if this needs to be further improved upon, in your opinion

  1. Median survival for PEL is stated as <1 year. In the paper by Lurain et al, median survival was 22 months [Lurain K et al. Viral, immunologic, and clinical features of primary effusion lymphoma. Blood. 2019 Apr 18;133(16):1753-1761. doi: 10.1182/blood-2019-01-893339. Epub 2019 Feb 19. PMID: 30782610; PMCID: PMC6473499.]

Suggest providing a range of median survival to reflect these more recent improvements in outcomes. Both in table 1 and in section 5 on Prognosis and treatment (line 240).

Response: Thank you for bringing our attention to this detail, please see updated median survival range in table 1 and section 5 on Prognosis and Treatment, more accurately reflecting reference #3.

  1. Suggest adding another row to show association with other immunosuppressive conditions in HIV-negative individuals, especially for PEL and PBL.

Response: This is an excellent idea, thank you. We have incorporated these immunosuppressive and other associated conditions to each column of Table 1.

Please note that the responses below were not specifically numbered but have been addressed as your team suggested:

In discussing the differential diagnosis of PEL in patients with HIV, it would be important to include that compared to DLBCL, PEL is significantly associated with hypoalbuminemia, thrombocytopenia, and elevated IL-10 levels [Lurain K et al. Viral, immunologic, and clinical features of primary effusion lymphoma. Blood. 2019 Apr 18;133(16):1753-1761. doi: 10.1182/blood-2019-01-893339. Epub 2019 Feb 19. PMID: 30782610; PMCID: PMC6473499.]

Response: We appreciate you bringing more of our attention to this detail, and we have included this important fact in the updated manuscript, lines 339-340.

In discussing prognosis, would also add that among patients with PEL, EBV positivity is associated with improved survival whereas elevated IL-6 levels are associated with inferior survival [Lurain K et al. Viral, immunologic, and clinical features of primary effusion lymphoma. Blood. 2019 Apr 18;133(16):1753-1761. doi: 10.1182/blood-2019-01-893339. Epub 2019 Feb 19. PMID: 30782610; PMCID: PMC6473499.]. Additionally, would also explain that with concurrent MCD the presence of PEL and KS, would impact the overall survival of patients (Ramaswami R, Lurain K, https://ashpublications.org/bloodadvances/article/5/6/1660/475480/Characteristics-and-outcomes-of-KSHV-associated).

Response: Thank you so much for this expansion of detail – we have updated the sections related to prognosis and HIV-positive PEL (lines 361 and 409, respectively) to reflect this important data, as well as the addition of Dr. Ramaswami’s work to our reference list.

In discussing treatment options, the authors mention the use of rituximab but mention that it has been explored for use in CD20+ PEL. However, rituximab has been used in PEL regardless of cell surface expression of CD20. Suggest that the reviewers expand the discussion of use of rituximab and explore the rationale in greater detail. For example, in the study of EPOCH-R2 for PEL, investigators justify the inclusion of rituximab by applying the current understanding of the pathobiology of IL-6 syndromes in KSHV-MCD and KSHV-NHL. Specifically, rituximab is effective in KSHV-MCD which is CD20 negative; PEL patients have similar clinical symptoms, suggesting a common pathophysiology likely driven by KSHV-infected B cells associated with increases in inflammatory cytokines (IL-6, IL-10, and viral IL-6). It is hypothesized that by depleting depletes B cells, Rituximab may therefore reduce levels of inflammatory cytokines that would otherwise lead to the clinical features of PEL [REF: Lurain K, Ramaswami R et al. Phase I/II Study of Lenalidomide Combined with DA-EPOCH and Rituximab (DA-EPOCH-R2) in Primary Effusion Lymphoma in Patients with or with-out HIV. Blood. 2019;134(Supplement_1):4096-4096. doi:10.1182/blood-2019-122070].  Moreover, the authors fail to mention the common overlap of PEL with KS and KSHV-MCD. These overlap syndromes are distinct entities which call for more nuanced treatment approaches e.g. addition of rituximab in PEL/KSHV-MCD overlap (as noted above).

Response: This feedback was absolutely essential and you are absolutely correct, thank you so much – we have expanded our treatment section by including your suggestions with existing as well as a few new references. Please see insertion to the “Treatment” section below (line 436):

5.4   Overlap of PEL with Kaposi Sarcoma associated Herpesvirus Multicentric Castleman Disease (KSHV-MCD)

Kaposi sarcoma associated herpesvirus (HHV8) is directly associated with development of concurrent PEL and/or Multicentric Castleman Disease. This is likely due to a common viral etiology and KSHV’s ability to utilize B-cells as a persistent reservoir of inflammatory cytokines, leading to proliferation and transformation to PEL and MCD. Interestingly, EBV-positive status in PEL is associated with improved survival compared to the more commonly occurring HIV-DLBCL. However, higher than normal circulating inflammatory markers (specifically IL-6, IL-10, and viral IL-6) as well as increased HHV8 viral loads are often found in patients with concurrent PEL and KSHV-MCD, which confer a worse prognosis. [75,89].  NCCN guidelines recommend treatment of KSHV-MCD with rituximab-based regimens, however, it is recommended that patients with concurrent PEL receive cytotoxic chemotherapy in addition to these rituximab-based regimens [3,82,86]. Rituximab has been shown to downregulate inflammatory cytokines as well as work synergistically with common topoisomerase inhibitors and other antineoplastic agents [87, 88]. The effective use of rituximab regardless of CD20 status in patients with concurrent PEL and KSHV-MCD was evidenced by the work of Lurain et al who suggest that use of rituximab could decrease the likelihood of patients diagnosed with KSHV-MCV to develop PEL, given its downregulatory effects on inflammatory cytokines [75].

The authors also fail to include the emerging use of immunotherapy in HIV positive patients with cancer, including non-Hodgkin lymphoma. In the study by Uldrick et al. [Uldrick TS, Gonçalves PH et al. Assessment of the Safety of Pembrolizumab in Patients With HIV and Advanced Cancer-A Phase 1 Study. JAMA Oncol. 2019 Sep 1;5(9):1332-1339. doi: 10.1001/jamaoncol.2019.2244. PMID: 31154457; PMCID: PMC6547135], investigators showed that the use of pembrolizumab is safe and has clinically beneficial activity against PEL.

Response: Thank you for your insights! We have added this valuable reference to section 5.3, line 429 and our reference list has been updated appropriately.

The authors also do not mention the use of immunotherapy combined with immunomodulatory drugs as a viable treatment option for patients with relapsed/refractory PEL. In another study from the NCI, the use of pembrolizumab with or without pomalidomide also showed responses/disease control in patients with PEL [Lurain K, Ramaswami R, et al. Use of pembrolizumab with or without pomalidomide in HIV-associated non-Hodgkin's lymphoma. J Immunother Cancer. 2021 Feb;9(2):e002097. doi: 10.1136/jitc-2020-002097. PMID: 33608378; PMCID: PMC7898875.]

Response: We appreciate the addition of this reference, as the information contained would be extremely valuable to bring to the attention of the public. Please note the updated inclusion on line 404, as well as the addition of this reference.

Finally, in discussing future opportunities for treatment (line 339), the authors fail to mention targeting CD38, a common marker of PEL cells. Shah et al. successfully treated a patient with PEL using daratumumab, an anti-CD38 monoclonal antibody [Shah NN et al. Daratumumab in Primary Effusion Lymphoma. N Engl J Med. 2018 Aug 16;379(7):689-690. doi: 10.1056/NEJMc1806295. PMID: 30110586.]

Response: We agree that the section on future treatment directions needed to be expanded, thank you for this inclusion. We have updated the section describing future directions in treatment, as well as our reference section.

Reviewer 4 Report

The authors present a review of primary effusion lymphoma (PEL), a rare lymphoma subtype that is seen in patients with immunocompromised status, and outline the epidemiology, pathogenesis, diagnostic testing including differential diagnoses, and treatment options in great detail.

Overall the paper is comprehensive and covers the breadth of the topic nicely. I only have a couple of minor comments for the authors’ consideration.

Comments:

  1. Please remove the references in the abstract. 
  2. Although the authors have outlined the treatment modalities in the manuscript, I would recommend the authors to provide an algorithm/ approach to newly diagnosed patients with PEL, which would immensely help the readers.

Author Response

The authors present a review of primary effusion lymphoma (PEL), a rare lymphoma subtype that is seen in patients with immunocompromised status, and outline the epidemiology, pathogenesis, diagnostic testing including differential diagnoses, and treatment options in great detail.

Overall the paper is comprehensive and covers the breadth of the topic nicely. I only have a couple of minor comments for the authors’ consideration.

Comments:

  1. Please remove the references in the abstract.
  2. Although the authors have outlined the treatment modalities in the manuscript, I would recommend the authors to provide an algorithm/ approach to newly diagnosed patients with PEL, which would immensely help the readers.

Response: Thank you so much for your feedback, we appreciate your time. Please note that in response to your advice, references have been removed from the abstract and a proposed algorithm has been added and designated as Figure 2. This figure can be located in the manuscript at the end of section 5 and has been reproduced below.

Figure 2.

DA-EPOCH: dose adjusted etoposide, prednisone, vincristine, cyclophosphamide, doxorubicin

R-CHOP: rituximab and cyclophosphamide, doxorubicin, vincristine, prednisone

CHOP + HD-MTX: cyclophosphamide, doxorubicin, vincristine, prednisone and high dose methotrexate

Axi-Cel: axicabtagene ciloleucel (CAR-T cell therapy)

EPOCH-R: etoposide, prednisone, vincristine, cyclophosphamide, doxorubicin with rituximab

HDT/ASCR: high-dose therapy and autologous stem cell rescue

ESHAP: etoposide, methylprednisolone, high-dose cytarabine, cisplatin

Please note that items listed below were corrections suggested by the review team that have been directly addressed in the manuscript, but not explicitly attributed to independent reviewers.

  1. Please confirm who is the corresponding author

Response: Thank you and apologies for the lack of clarity. The corresponding author is Diamone A. Gathers.

  1. Please add (highlighted portions)

Response: We appreciate your attention to this oversight, please see updated location information below:

  1. Department of Internal Medicine, University of New Mexico, Albuquerque, 87131, NM, USA
  2. University of Arizona College of Medicine, Phoenix, 85721, AZ, USA
  3. Division of Hematopathology, Department of Laboratory Medicine and Pathology, Mayo Clinic, Phoenix, 85054, AZ, USA
  4. Division of Hematology and Oncology, Mayo Clinic, Phoenix, 85054, AZ, USA

  1. According to our rules: A maximum of two joint first authors can be indicated by the addition of a superscript symbol. The symbol must be defined below the affiliations by addition of the following statement “These authors contributed equally to this work”.

Response: Thank you for reminding us of this detail, authorship has been updated.

  1. Please extend the simple summary to about 100 words

Response: Thank you for your comments. This section has been revised to a length of 139 words, which can hopefully more appropriately encompass our goals in composing this work. Please see the revision updated below, now spanning lines 15-25 of the revised manuscript:

Primary effusion lymphoma (PEL) is a rare B-cell lymphoma with a particularly aggressive course. This disease usually affects the variably immunocompromised and maintains a predilection for body cavities, leading to the development of malignant effusions. Pathogenesis suggests an association with HHV8 and EBV, although this is unclear. There are no definitive guidelines for the treatment of PEL but commonly used initial regimens include those similar to DA-EPOCH and CHOP. The prognosis remains poor, even with treatment. Newer therapies incorporate the use of axicabtagene ciloleucel, CAR-T cell therapies and augmented EPOCH regimens. Further investigation into oncogenesis and targeted molecular pathways could provide insights into treatment targets. In this review, the authors would like to compare this disease with other similarly aggressive B-cell lymphomas as well as highlight the unique epidemiology, pathogenesis and current treatment options for patients diagnosed with PEL.

  1. Please confirm whether the figures in your manuscript were made by authors. If any images were copied or adapted from other papers, please send the copyright permission to us (Figure 1).

Response: Thank you for your attention to this detail. Please note that Figure 1 is an originally created image, serving as an expanded perspective on the combined work of Shimada, Hayakawa and Kiyoi who described the biology and management of primary effusion lymphoma in 2018. There does not appear to be a copyright permission, however we have referenced the original authors in our reference list as below:

Reference 4: Shimada K, Hayakawa F, Kiyoi H. Biology and management of primary effusion lymphoma. Blood. 11 2018;132(18):1879-1888. doi:10.1182/blood-2018-03-791426

  1. We compared your paper with the publications and find there are some overlaps. Please re-write the highlighted parts in your own words

Response: We greatly appreciate your insights and would like to modify the aforementioned content, previously spanning lines 67-132 below, updated to lines 92-160:

The key for establishing a persistent HHV8 infection in the human cell is a complex interaction between HHV8 and the host immune system. HHV8 accomplishes this task by producing proteins homologous to human immune modulatory proteins, that have been likely hijacked from human host cells over the course of viral evolution [4]. Persistent infections can be established by either a latent or a lytic viral life cycle. The latent life cycle results in a prolonged, stable, immunologically silent infection. During the lytic cycle virions are released and infect new host cells [4, 17]. HHV8 engages sophisticated cellular processes to make a persistent infection possible by interfering with host immune responses.

A latent infection specifically targeting lymphoid cells is not unique to HHV8 but is a general feature of gamma herpesviruses [17]. Most lymphoproliferative disorders develop many years after a primary viral infection by a gamma herpesvirus, underlining the medical importance of the persistent latent infection. It is known that lymphoma cells of PEL are in the latent phase of infection, a viral life cycle that is characterized by lack of expression of most viral antigens [4, 17]. While many other proteins are not expressed in latency, abundant expression of the latency-associated nuclear antigen (LANA) is crucial for the preservation of the viral genome in host cells during replication [19]. Why are potential vi-ral epitopes on LANA are not provoking a strong response by the cytotoxic T-cells (CTLs) [20,21]. One of the main immune-escape mechanisms of the latency phase is the inhibition of CTLs, a feat achieved by engaging a variety of proteins produced by HHV8, including viral caspase-8 and (FLICE)-like inhibitory protein (vFLIP). Viral FLIP is a protein of strong anti-apoptotic properties in T-cell lines [22], and a promoter of nuclear factor kB (NF-kB) in host cells through engaging members of the TRAF family (tumor necrosis factor receptor-associated factor family) [23, 24].

Many proteins produced by HHV8 target a central paradigm of T-cell mediated immune regulation, notably, a promotion of the host TH2-cell responses and concomitant downregulation of the TH1-dependent T-cell immunity. Several virally encoded cytokines, especially viral interleukin-6 (vIL-6), viral chemokines, and CD200 play a central role in this process. The pleiotropic cytokine IL-6 exhibits complex roles in human cells with important effects in hematopoiesis, inflammation, and oncogenesis by binding to one of two receptors, IL-6 receptor, and gp130. HHV8 exploits these interactions by producing a homologue of IL-6 (vIL-6) that is encoded by the open-reading frame (ORF) K2 of the HHV8 genome. The expression of vIL-6 is well documented in both lytically infected as well as latently infected cells [25, 26]. An important feature of vIL-6 is the capability to activate gp130 independently of the IL-6 receptor [27, 28]. Once gp130 is activated, vIL-6 can initiate different pathways of immune evasion. Downstream effects of vIL-6 include the promotion of TH2-cell response over a TH1-type response [30], and the inhibition of influx of neutrophils by increasing expression of the chemokine CCL2 and decreasing expression of CXCL8 [29].

Viral IL-6 is not the only human cytokine homologue produced by HHV8. Viral CC-chemokine ligand 1 (cCCL1), vCCL2, and vCCL3, which are also encoded by the HHV8 genome [25], target both TH2-cells and CD4+CD25+ regulatory T-cells [31]. ORF K14 of HHV8 encodes viral CD200 (vCD200), a glycosylated cell surface protein with a strong inhibitory effect on TH1-type cytokine production [32]. HHV8 also produces 4 different interferon (IFN)-regulatory factors viral IRF1, vIRF2, vIRF3 and vIRF4, which are produced during lytic reactivation and latency phase. Like their human homologues, these regulatory proteins inhibit type I interferon (IFN)-mediated signaling and IFN-mediated apoptosis [33].

In addition to its interference with the cell-mediated immunity, HHV8 also modulates the activity of the complement system. The complement system is capable to link the innate and adaptive immune systems and plays an important role against viruses. HHV8 evades complement-mediated immunity by producing three complement lytic proteins (KCPs) that are encoded by ORF4 and are produced by alternative splicing [34]. KCPs con-tribute the lysis of C3 convertase, the main product of the classical complement pathway and they can inactivate C3b and C4b. Once intact C3b and C4b are bound to the infected cell, they could exhibit a strong anti-viral effect by either lysing an infected cell or coating the viral particle and promoting complement-mediated phagocytosis [35]. The importance of this immune evasion mechanism is lymphoma genesis is highlighted by the observation of KCPs detected on the surface of HHV8-infected cells of PEL [34].

Immune eradication of viral infections requires a strong CD8+ CTL mediated attack on virus-infected cells. These cytotoxic cells are strongly dependent on MHC class I-dependent antigen presentation, therefore, they require a strong response by antigen presenting cells (APCs), such as dendritic cells, macrophages, and B cells. HHV8 engages several strategies to escape recognition by CD8+ CTLs. HHV8 infected cells express two homologous proteins, modulator of immune recognition 1 (MIR1) and MIR2, both capable of downregulating expression of MHC class I molecules on the surface of infected cells [36, 37, 38].  These two proteins have been shown to be expressed in PEL and in latently infect-ed B-cells [39]. An additional effect of vMIR2 is decreasing the expression of CD86 and ICAM1 resulting in the inhibition of APCs [40, 41].

  1. We compared your paper with the publications and find there are some overlaps. Please re-write the highlighted parts in your own words (previously lines 147-157).

Response: Thank you for bringing our attention to this area. We reviewed the literature and incorporated your suggestion and would like to update the language for this section, previously lines 147-157, now 243-252.

The diagnosis of PEL may be made by evaluation of a body fluid or a solid tissue mass in those cases with an extracavitary presentation. The neoplastic cells may be quite pleomorphic in body fluids, ranging from large immunoblastic/plasmablastic cells to more anaplastic-appearing cells (Figure 2) [1, 4, 44, 45, 46]. The lympho-ma cells contain large nuclei with round to irregular nuclear contours and prominent nucleoli, and generally contain abundant basophilic cytoplasm. Some cells may show a perinuclear “hof” or halo compatible with plasmacytoid differentiation. Occasional multinucleated Reed-Sternberg-like cells may be present. Although the morphologic features of extracavitary PELs are similar, the neoplastic cells may have a more uniform appear-ance than those found in body fluids, and such cases may show involvement of lymph node sinuses or other lymphatic or vascular channels mimicking anaplastic large cell lymphoma or intravascular lymphoma [44,45].

  1. Please confirm whether the figures in your manuscript were made by authors. If any images were copied or adapted from other papers, please send the copyright permission to us (Figure 2).

Response:  We greatly appreciate you bringing this oversight to our attention. Please note that Figure 2 is an original image, provided by one of our team pathologists and co-authors Dr. S. Gibson and that there is not associated copyright permission. Thank you.

  1. We compared your paper with the publications and find there are some overlaps. Please re-write the highlighted parts in your own words (previously lines 166-181).

Response: Thank you for your comments and we apologize for any oversight. This section (previously lines 166-181) has been revised and is now updated to span lines 278-290:

The neoplastic cells of PEL originate from post-germinal center B cells that show further plasmablastic differentiation [4, 44]. The cells usually express the leukocyte common antigen CD45 and show variable staining for plasma cell-associated markers, such as CD138, CD38, IRF4/MUM1, and EMA [1, 4, 44, 45, 46]. The neoplastic cells generally lack expression of the pan-B-cell markers CD19, CD20, CD79a, and PAX5, and surface and cytoplasmic immunoglobulin is usually negative or low in expression. However, it is noted that the extracavitary variant of PEL shows slightly more frequent expression of B-cell associated markers and immunoglobulins compared to classic PEL [44, 45, 46]. Although T-cell and natural killer cell-associated antigens are generally negative, aberrant expression of some T-cell markers, including CD3, may occasionally be seen, particularly in those cases with extracavitary presentation [4, 44, 45, 46]. CD30, which is frequently expressed by PELs, raises the differential diagnosis of both anaplastic large cell lymphoma and classic Hodgkin lymphoma [4, 44, 45, 46]. However, the diagnosis of PEL can be confirmed with immunohistochemical staining for LANA1, an HHV8-associated latent protein, which is positive in the nuclei of the PEL cells [4, 10, 44, 45, 46].  EBV may be demonstrated with in situ hybridization for EBV-encoded small RNA (EBER) in many cases, while EBV LMP1 is ab-sent [44, 46].

  1. Please confirm whether the table in your manuscript need copyright permission. If yes, please provide it to us. If not, please provide the explanation (Table 1).

Response: Thank you for your attention to this detail. Table 1 is an original table combined by the authors from several referenced pathology sources and our team pathologist. In response to your insights, the title has been updated from “Comparative Features of Aggressive Primary B-Cell Lymphomas” to “Differential Diagnosis of Primary Effusion Lymphoma” to better reflect our goals and considerations for the readers in preparing this table and we believe that there is not an associated copyright permission.

  1. We compared your paper with the publications and find there are some overlaps. Please re-write the highlighted parts in your own words (previously lines 309-325).

Response: Thank you for your attention to this detail. We revisited the literature and modified the language composing this section, now covering lines 422-440:

Axicabtagene ciloleucel (Axi-cel) is a chimeric antigen receptor (CAR) T-Cell agent, currently used as a third line treatment for follicular and large B-cell lymphoma. ZUMA-1 is a phase 2 clinical trial across multiple centers, including 111 patients. Axi-cel was successfully manufactured for 99% of these patients and administered to 91% resulting in an overall response rate (ORR) of 82% and a complete response rate (CRR) of 54%. These responses were sustained with a median follow-up of about 15 months and OS at 18 months of 52% [64].   Brexucabtagene autoleucel (previously KTE-X19) is another CAR-T cell therapy, typically used for relapsed/refractory mantle cell lymphoma.  This therapy was used in the 74-patient ZUMA-2 trial, ultimately showing a 12-month progression free survival (PFS) of 61% and OS of 83%. The most common adverse events were those effects previously experienced after CAR-T therapy, including cytopenias (94%) and infections (32%) [65].   Simi-lar success was observed in the smaller ATTCK-20-2 trial, which incorporated the use of ACTR087 plus rituximab in patients with relapsed/refractory CD20-positive B-cell lymphoma. Of the 26 patients enrolled, 69% were diagnosed with relapsed/refractory DLBCL and subsequently treated with fixed doses of rituximab administered concurrently with variable doses of ACTR087 leading to an ORR of 50%. In addition, the team was able to further define safety monitoring and adverse reaction management guidance across clinical studies inclusive of ACTR T-cell products [66].

  1. Please list first 10 authors before using “et al.”.

Response: We appreciate you bringing this oversight to our attention. Please note that we have carefully reviewed each reference in our list to ensure adherence to appropriate guidelines, as you’ve outlined. Examples of the updated format are listed below, with reference 3 having more than 10 authors and reference 13 having less than 10 authors:

Reference 3: Lurain K., Polizzotto MN, Aleman K., Bhutani, M., Wyvill, K., Goncalves, P., Ramswami, R., Marshall, V., Miley, W., Steinberg, S. et al Viral, immunologic, and clinical features of primary effusion lymphoma. Blood. 04 2019;133(16):1753-1761. doi:10.1182/blood-2019-01-893339

Reference 13: Pan ZG, Zhang QY, Lu ZB, Quinto, T., Rozenvald, I., Liu, L., Wilson, D., Reddy, V., Huang, Q., Wang, H., Ren, Y. Extracavitary KSHV-associated large B-Cell lymphoma: a distinct entity or a subtype of primary effusion lymphoma? Study of 9 cases and review of an additional 43 cases. Am J Surg Pathol. Aug 2012;36(8):1129-40. doi:10.1097/PAS.0b013e31825b38ec

Round 2

Reviewer 3 Report

No additional issues